# EnP1 exploits H2Aub-dependent epigenetic reprogramming to promote microsporidia proliferation in host cells

Jingyu Guan, Yongliang Wang, Ming Fu, Liyuan Tang, Musa Makongoro Sabi, Huimin Zhu, Hua Cong, Chunxue Zhou, Huaiyu Zhou, Hongnan Qu*, Bing Han *

Department of Pathogenic Biology, School of Basic Medical Sciences, Cheeloo College of Medicine, Shandong University, Jinan, Shandong, China

* quhn@sdu.edu.cn (HQ); bing.han@sdu.edu.cn (BH)

## Abstract

Microsporidia, as opportunistic parasitic pathogens, constitute a formidable threat to human health. Although the regulatory circuitry of the nucleus-targeted effector EnP1 remains highly intricate and only partially characterized, our study identifies histone H2A as a novel binding partner of EnP1. Furthermore, we demonstrate that both EnP1 overexpression and microsporidia infection induce monoubiquitination of H2A (H2Aub) through downregulation of BAP1 expression. Subsequent mechanistic analyses revealed that elevated H2Aub levels positively correlate with enhanced microsporidian proliferation, whereas attenuation of H2Aub markedly suppresses pathogen expansion. Furthermore, EnP1 orchestrates the enrichment of H2Aub at the *SLC7A11* promoter, driving its transcriptional upregulation. Collectively, these findings underscore that EnP1 modulates the ferroptosis state of host cells through H2Aub-mediated epigenetic reprogramming, ultimately facilitating pathogen propagation. This study endeavors to elucidate the critical survival strategies of microsporidia within host cells mediated by EnP1 and to unravel the multifaceted interplay between these pathogens and their hosts.

## Author summary

Microsporidia can induce multi-organ infections and tissue damage in humans, with severe cases posing life-threatening risks. EnP1, a validated nucleus-targeted effector of microsporidia, drives epigenetic reprogramming of host cells, although its regulatory network remains incompletely characterized. In this study, we confirmed the direct interaction between EnP1 and host histone H2A and demonstrated that microsporidia exploit the secretion of EnP1 during infection to suppress deubiquitinase BAP1 expression, thereby elevating H2Aub and establishing an intracellular environment conducive to pathogen proliferation. We

**Data availability statement:** Underlying data can be found via Guan, Jingyu; Han, Bing (2025), "EnP1 exploits H2Aub-dependent epigenetic reprogramming to promote microsporidia proliferation in host cells", Mendeley Data, V4, doi: 10.17632/bhscx3g5fc.4.

**Funding:** This work was supported by the National Natural Science Foundation of China (824B2068 to JG), Guangdong Basic and Applied Basic Research Foundation (2024A1515010685 to BH), the National Natural Science Foundation of China (82572601 to BH; 82502750 to HQ), Guangdong Basic and Applied Basic Research Foundation (2023A1515110681 to QH), Shandong Provincial Natural Science Foundation (ZR2024QH030 to BH; ZR2023QC236 to QH), Shenzhen Fundamental Research Program (JCYJ20250604124228036 to BH), the QILU Young Scholars Program of Shandong University (215510082063092 to BH). The funders had no role in study design, data collection and analysis, decision to publish, or preparation of the manuscript.

**Competing interests:** The authors have declared that no competing interests exist.

further revealed that EnP1 orchestrates H2Aub enrichment at the *SLC7A11* promoter region, enhancing its transcriptional activation and subsequently inhibiting ferroptosis in host cells. These findings systematically expand the interactome of EnP1 and its anti-ferroptotic regulatory framework while providing definitive evidence for the functional pleiotropy of microsporidia effectors.

## Introduction

The interplay between pathogens and their hosts is multifaceted and highly intricate, with pathogens counteracting host immune defenses by orchestrating epigenetic alterations within host cells to regulate the expression of crucial genes [1]. Epigenetic modifications, which govern gene expression without modifying the underlying genetic sequence, represent a pivotal mechanism through which host cells adapt to pathogenic infections and reprogram their transcriptional activity [2,3]. Effectors secreted by pathogens exert extensive influence on host cellular functions by engaging with specific molecular targets within the host cells [4]. Notably, nuclear-localized effectors can instigate dysregulated gene expression by orchestrating multifaceted modulation of epigenetic alterations [5]. The microsporidian effector EnP1, which targets the host cell nucleus, possesses the capacity to suppress ferroptosis by modulating histone modifications that perturb downstream gene expression; however, the precise underlying mechanism remains incompletely understood [6].

Microsporidia, as obligate intracellular parasites with broad host tropism encompassing virtually all animal species, are etiological agents of microsporidiosis, a clinically significant zoonosis [7,8]. These pathogens pose substantial threats to both public health and economic sectors, particularly impacting aquaculture and sericulture industries [9,10]. Pathogenic manifestations include multi-organ infections affecting the gastrointestinal tract, ocular system, and central nervous system, with microsporidiosis representing a major opportunistic infection contributing to AIDS-related mortality [8,11]. Of particular concern is the escalating incidence of ocular infections in immunocompetent individuals, notably manifesting as microsporidial keratoconjunctivitis with markedly increased prevalence [12]. The primary transmission occurs via fecal-oral route, with demonstrated potential to instigate waterborne outbreaks [13,14].

Microsporidia are exceptionally specialized intracellular parasites, exhibiting an absolute dependence on their host for both nourishment and survival [15]. Through the course of evolution, its genome has undergone significant reduction, enabling it to manipulate the host cell's intracellular homeostasis to promote its proliferation and dissemination [8,16]. The spore wall protein EnP1, implicated in microsporidian infestation, exhibits a unique role as a secreted protein targeting the host cell nucleus, thereby highlighting the functional versatility of key proteins in microsporidia despite their genome shrinkage [6,17].

The regulation of ferroptosis in host cells by pathogens is a multifaceted and intriguing process, which, in most cases, appears to facilitate pathogen proliferation

[18]. Microsporidia effectively inhibit host cell apoptosis [19,20], obstruct cell cycle progression [21], and suppress host immune responses [22]. However, their regulation of host cell ferroptosis remains inadequately explored. Histones H2A and H2B form a heterodimer through hydrophobic interactions and salt bridges, which is essential for nucleosome assembly and chromatin organization [23]. The extent of host cell H2Bub plays a pivotal role in regulating ferroptosis induced by microsporidia, while the precise function of H2A as a structural partner for H2B remains elusive [6]. A multitude of studies have demonstrated that H2Aub (K119) and H2Bub (K120) can collaboratively engage in host cell defense mechanisms during pathogen invasion, including the regulation of host cell iron homeostasis [24,25]. Therefore, the impact of H2A and its post-translational modifications on host ferroptosis in the context of microsporidian infection warrants further investigation and elucidation.

In this study, we explored the mechanism by which the effector EnP1 inhibits ferroptosis by modulating host cell H2Aub levels. We confirmed the presence of H2A, a reciprocal target of EnP1, in host cells and demonstrated that its H2Aub levels are elevated through the downregulation of the deubiquitinating enzyme BAP1. Notably, the increased H2Aub levels, triggered by EnP1 or *Encephalitozoon hellem* (Eh) infection, facilitated the upregulation of SLC7A11 by enhancing the binding affinity of H2Aub to the *SLC7A11* promoter, representing a key mechanism through which microsporidia counteract host cell ferroptosis during infection. This work enriches our understanding of the regulatory strategies employed by EnP1 in modulating host cell ferroptosis, contributing to a deeper insight into the pathogenesis of microsporidia and offering valuable clues and therapeutic targets for the development of novel anti-infective therapies. Such findings hold significant implications for safeguarding human health and related economic sectors, including aquaculture.

## Results

### EnP1 forms a complex with host H2A

Pathogen-secreted effectors have the capacity to interact with multiple intracellular targets in host cells, thereby disrupting cellular functions and resulting in the dysregulation of gene expression [26]. EnP1 has previously been identified as a critical nucleus-targeted effector [6]. To further elucidate the host epigenetic regulatory network influenced by EnP1 through its interaction with host nucleus proteins (S1A Fig), we established a stable heterologous expression of EnP1-HA (S1B Fig), and performed proteomic analysis on EnP1-HA co-immunoprecipitation (Co-IP) to identify EnP1 interacting proteins. Silver staining of the EnP1 Co-IP complexes revealed significantly enriched protein bands in the EnP1 group versus the control (Fig 1A). Immunoblot analysis identified H2A in the EnP1 Co-IP complexes, indicating H2A as a potential interaction partner of EnP1 and further validating our previously published mass spectrometry data [6], which designated H2A as the predominant interactor (Fig 1B, S1 Table).

To validate the EnP1-H2A interaction, we predicted their complex structure using AlphaFold3 [27]. Although the modeling identified potential interaction interfaces, the low prediction confidence (pLDDT) of EnP1 limits the reliability of the EnP1-H2A structural model, suggesting it should be interpreted with caution (S1C and S1D Fig). Initial fluorescence colocalization suggested potential EnP1-H2A interaction (Fig 1C), which was subsequently confirmed by His pull-down assays using co-expressed EnP1-HA-His and H2A-Flag constructs, demonstrating direct complex formation (Figs 1D, S1E and S1F). Furthermore, Co-IP experiments confirmed the reciprocal association between EnP1 and H2A (Fig 1E and 1F).

Collectively, these results suggest that H2A is another crucial interaction target of the microsporidian effector EnP1 in the host cell nucleus. This finding not only reveals the functional complexity of EnP1 within the framework of its highly reduced genome but also implies the potential regulatory function of EnP1 in modulating the host through H2A.

### EnP1 enhances the monoubiquitination of H2A by downregulating BAP1 expression

H2A and H2B, existing as dimers on either side of the nucleosome, collaboratively mediate nuclear homeostasis through post-translational modifications and interactions with histone variants [28]. The monoubiquitination of histone

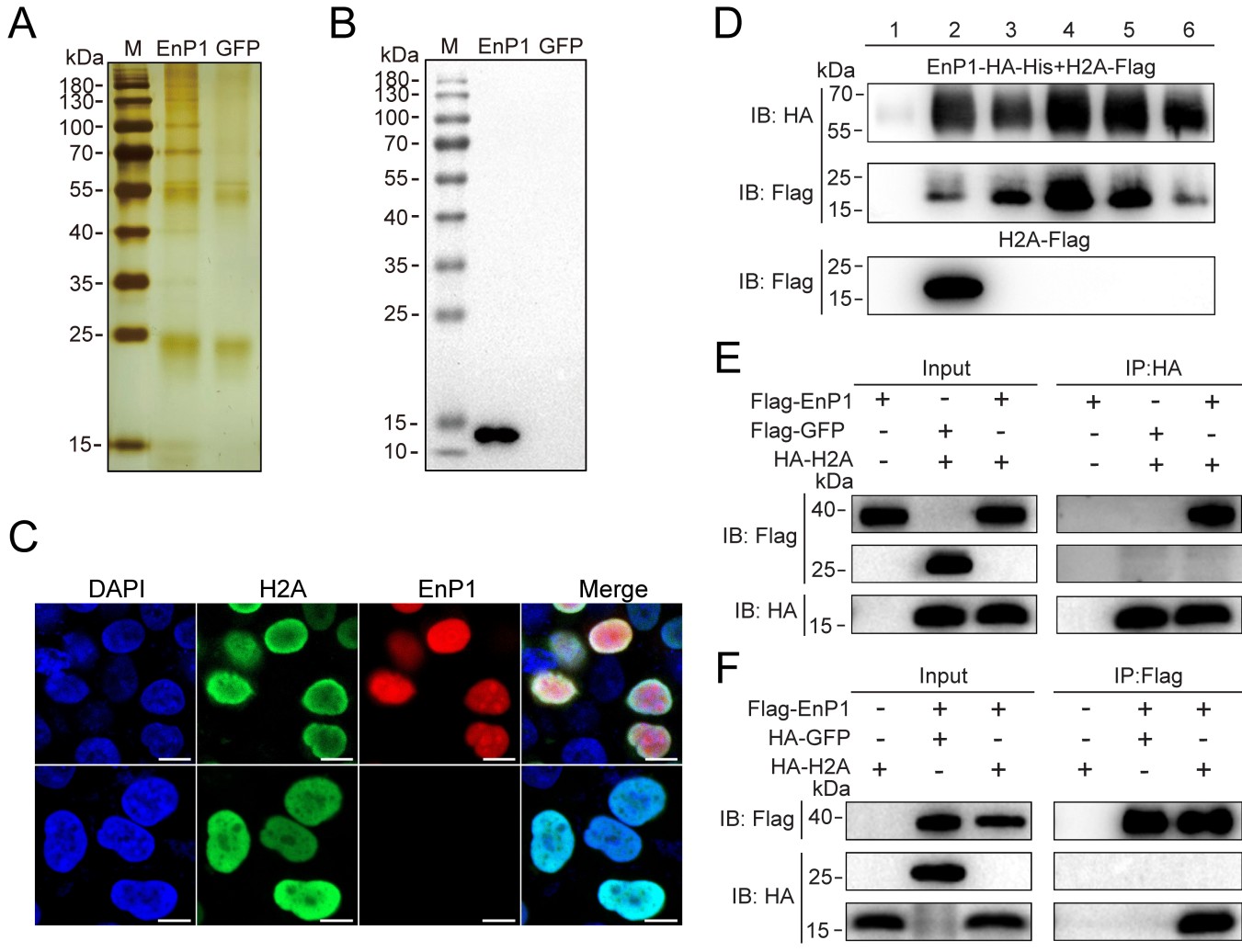

**Fig 1. EnP1 interacts with host H2A. (A)** Silver staining of anti-HA immunoprecipitates from EnP1-HA stable cells, with GFP stable cells as the negative control. **(B)** Immunoblot analysis of histone H2A in EnP1 IP complexes. Negative control: GFP stable cells. **(C)** Fluorescence co-localization analysis was performed in HEK293T cells co-transfected with pcDNA3.1::EnP1-Flag and pcDNA3.1-3HA::H2A constructs. Nuclear co-localization of EnP1 and H2A was visualized using α-Flag (red) and α-HA (green) mAb, respectively. The merged image confirms nuclear co-localization of EnP1 and H2A (Scale bar: 20 μm). **(D)** Immunoblot analysis of prokaryotic co-expression and purification of H2A and EnP1. The pRSFDuet prokaryotic expression vector was utilized for the co-expression of EnP1-HA-His and H2A-Flag. Samples 1 and 2 represent uninduced and post-induced bacterial lysates, while samples 3 to 6 show proteins sequentially eluted with 500 mM imidazole, demonstrating that the EnP1 protein can pull down H2A without the His-tag. **(E)** Immunoblot analysis of nuclear proteins co-expressed with H2A and EnP1, using anti-HA magnetic beads to immunoprecipitate proteins from HEK293T cell nuclear extracts. **(F)** Immunoblot analysis following immunoprecipitation of H2A and EnP1 co-expressed nuclear proteins from HEK293T cells using anti-Flag magnetic beads.

H2A, as a pivotal participant in epigenetic regulation, can synergize with H2Bub to play a crucial role during pathogen invasion of the host [25,29]. Thus, we hypothesize that during infection, microsporidia may modulate the host cell H2Aub levels through the nuclear translocation of EnP1 to facilitate its growth. To validate this hypothesis, we utilized a modification-specific antibody against H2Aub (K119) and demonstrated that both EnP1 overexpression (Fig 2A and 2B) and microsporidia infection (Fig 2C and 2D) significantly elevated H2Aub levels in host cells, independent of total H2A protein abundance (S2A and S2B Fig). To exclude potential confounding effects from other H2A post-translational

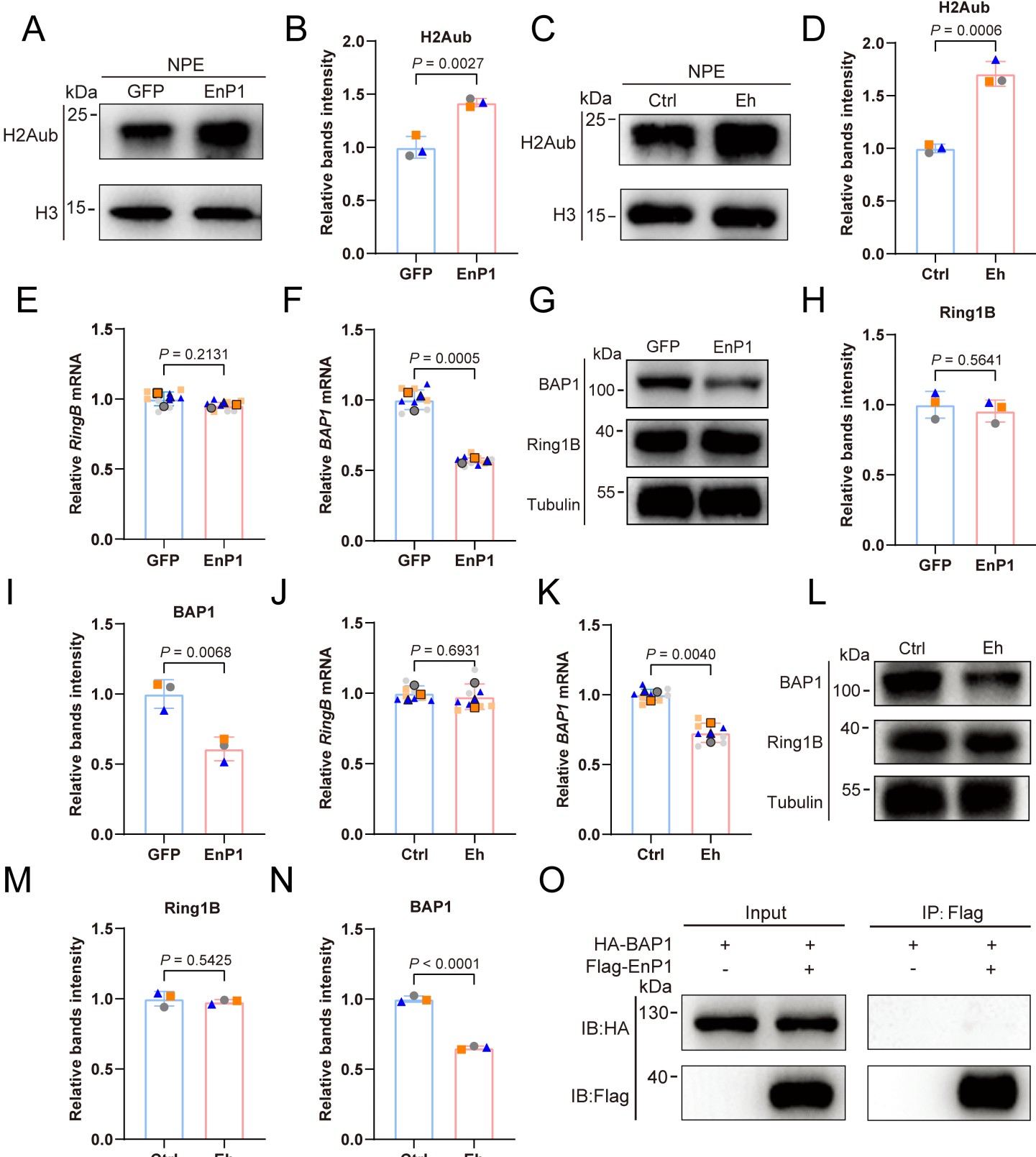

**Fig 2. EnP1 promotes H2Aub modification in host cells by restraining BAP1 expression. (A-B)** Immunoblotting analysis of H2Aub levels in EnP1-stabilized cells (A) and quantification of relative band intensities from three independent biological replicates using ImageJ software (B). **(C-D)**

Immunoblot analysis of H2Aub levels in host cells infected with microsporidia (C) and quantification of relative band intensities from three independent biological replicates using ImageJ software (D). NPE: nuclear protein extractions. **(E)** qRT-PCR analysis of the alteration in RING1B expression in host cells stably expressing EnP1. **(F)** qRT-PCR analysis of the alteration in BAP1 expression in host cells stably expressing EnP1. **(G)** Immunoblot analysis of the alteration in BAP1 and RING1B expression in host cells stably expressing EnP1. **(H-I)** Three independent biological replicates of RING1B (H) and BAP1 (I) protein expression levels in host cells stably expressing EnP1 were analyzed by band intensities. **(J)** qRT-PCR analysis of the change in RING1B expression in host cells infected with microsporidia. **(K)** qRT-PCR analysis of the alteration in BAP1 expression in host cells infected with microsporidia. **(L)** Immunoblot analysis of the alteration in BAP1 and RING1B expression in host cells infected with microsporidia. **(M-N)** Relative band intensities of three biological replicates of RING1B (M) and BAP1 (N) protein expression levels in host cells infected with microsporidia. **(O)** Immunoblot analysis following immunoprecipitation of NPE from HEK293T cells co-expressing BAP1 and EnP1 using anti-Flag magnetic beads. Quantification of mRNA and protein expression for each target was performed with three biologically independent replicates to ensure experimental reproducibility.

modifications (PTMs), we confirmed that neither H2AK5ac nor H2AK9ac levels were altered by EnP1 expression or pathogen infection (S2C-S2F Fig).

H2A monoubiquitination is mediated by the Polycomb repressive complex 1 (PRC1), which contains the E3 ubiquitin ligases RING1A and RING1B [30]. RING1A mainly functions as a regulatory subunit, while RING1B acts as the primary catalytic enzyme that determines histone H2Aub levels [31]. Conversely, the removal of H2Aub, or deubiquitination, is primarily mediated by the Polycomb repressive deubiquitinase (PR-DUB) complex, with BAP1 as its core deubiquitinating enzyme [32,33]. Other enzymes, such as USP16 can also moderately influence H2Aub levels [34,35]. To further elucidate the mechanism by which EnP1 regulates H2Aub in host cells, we evaluated the expression of key modifying enzymes involved in H2A modification in both EnP1 stable cells and microsporidia-infected cells. Our results show that the transcriptional or protein levels of the RING1A (S2G-S2L Fig) and RING1B (Fig 2E and 2G-2H, 2J and 2L-2M) remained unchanged under these conditions, suggesting that the ubiquitination of H2A does not result from altered PRC1 activity. On the other hand, while USP16 levels remained stable (S2M-S2R Fig), we observed a marked downregulation of both mRNA and protein levels of the BAP1 in response to EnP1 expression (Fig 2F-2G and 2I) and microsporidia infection (Fig 2K-2L and 2N). To further investigate the potential mechanism of BAP1 suppression, we performed co-immunoprecipitation (Co-IP) assays, which demonstrated that BAP1 does not physically interact with EnP1 (Fig 2O). These rules out the possibility that EnP1 directly binds to and modulates BAP1 activity. These findings collectively indicate that EnP1 regulates H2Aub accumulation not by altering the activity of PRC1, but primarily through its targeted suppression of host BAP1, thus preventing the deubiquitination of H2A.

## Host cell H2Aub elevation enhances microsporidian growth

The modification of histone H2Aub meticulously govern the downstream gene expression in host cells and are intricately linked to the cellular state of the host [25]. However, the specific impact of H2Aub alteration, induced by the microsporidian effector EnP1, on the proliferation of microsporidia remains unclear. To investigate this, we assessed microsporidian proliferation by measuring pathogen load at 0 and 12 hours post-infection (hpi) using absolute quantitative qPCR, which served both to evaluate washing efficiency and methodological feasibility (S3A and S3B Fig).

To explore the role of H2Aub in microsporidian growth, we overexpressed BAP1 in host cells, which successfully reduced H2Aub modification levels (Figs 3A and S3C). Importantly, this approach did not impact cellular viability (S3D and S3E Fig), thereby eliminating potential interference with microsporidia proliferation assays. A significant reduction in H2Aub levels was found to markedly suppress microsporidia proliferation. This inhibitory effect was demonstrated by a pronounced decrease in pathogen load at 48 hpi (Fig 3B), along with a substantial reduction in the number of parasitophorous vacuoles (PVs) (Fig 3C) and their area size (Fig 3D). Conversely, knockdown of BAP1 expression in host cells (S3F and S3G Fig) led to an increase in H2Aub levels (Fig 3E). Under conditions where cellular viability remained unaffected (S3H and S3I Fig), the proliferative capacity of microsporidia was significantly enhanced (Fig 3F–3H).

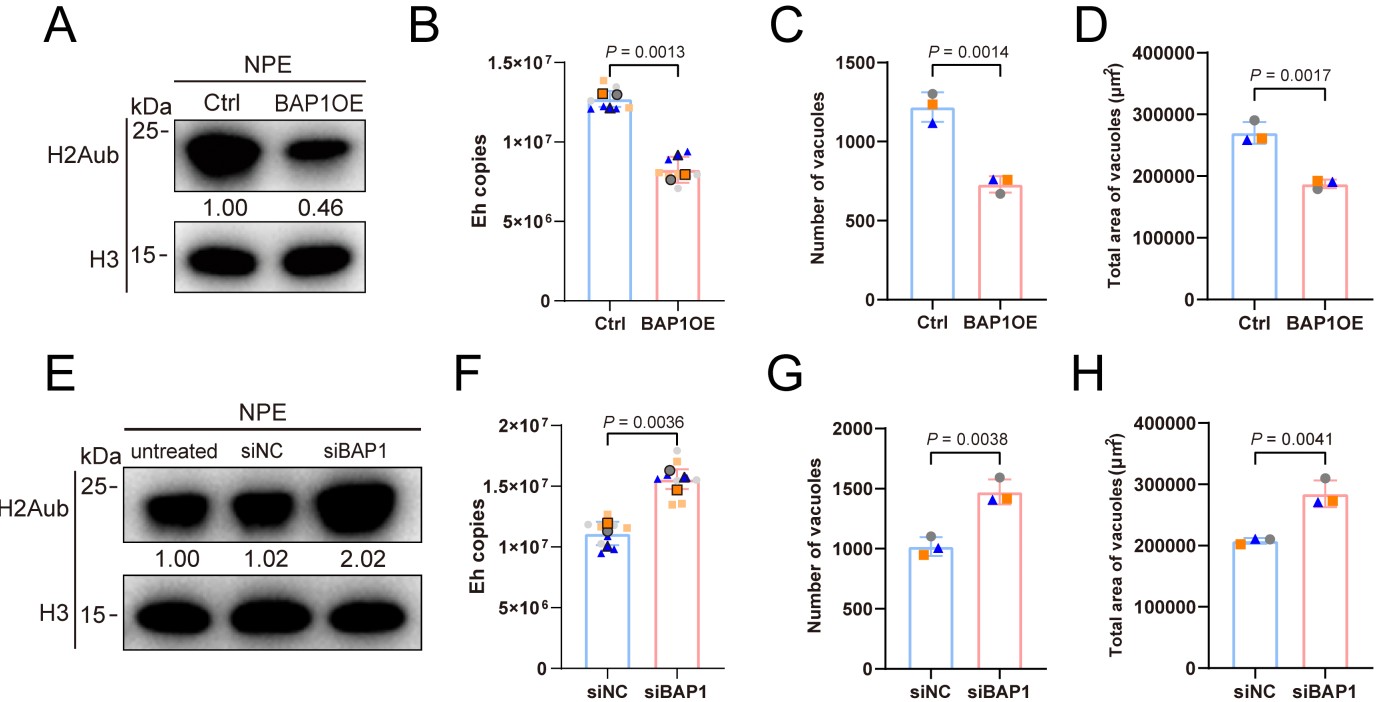

**Fig 3. The level of H2Aub in host cells is positively correlated with microsporidia proliferation. (A)** Immunoblot analysis of H2Aub levels in host cells after overexpressing BAP1. **(B)** Parasite load of microsporidia in HEK293T cells overexpressing BAP1 at 48 hours post-infection (hpi), with pathogen copy number determined by quantitative PCR. Multiplicity of infection (MOI)=10. **(C-D)** The effect of BAP1 overexpression in host cells on microsporidia infection was evaluated by counting the number of PVs on each slide (C) and the total area of PVs within the infected cells (D) at 48 hpi. MOI = 10. **(E)** Immunoblot analysis of H2Aub levels in host cells following BAP1 knockdown. siNC: A non-targeting siRNA with scrambled sequence, serving as a negative control to rule out off-target effects. **(F)** Parasite load of microsporidia in HEK293T cells overexpressing BAP1 at 48 hpi, with pathogen copy number determined by quantitative PCR. MOI = 10. **(G-H)** The effect of BAP1 knockdown in host cells on microsporidia infection was evaluated by counting the number of PVs on each slide (G) and the total area of PVs within the infected cells (H) at 48 hpi. MOI = 10. Microsporidia parasite load, PVs counts, and PVs areas were analyzed across three biologically independent replicates.

This phenomenon mirrors the positive regulation of host cell H2Aub levels by EnP1, the secreted effector of microsporidia, which also promotes microsporidian proliferation. Taken together, these findings demonstrate that an elevation of host cell H2Aub significantly promotes the proliferative ability of microsporidia within the host cells.

## EnP1 promotes SLC7A11 expression by enhancing the binding of H2Aub to the *SLC7A11* promoter

In our previous study, we demonstrated that EnP1 significantly upregulates the expression of the ferroptosis inhibitor SLC7A11, by interacting with histone H2B and disrupting its ubiquitination (H2Bub) [6]. Given that H2A and H2B form a heterodimer, a critical structural unit of the nucleosome, we hypothesized that H2A might also be involved in EnP1-mediated regulation of host cell processes. In addition, previous reports have shown that SLC7A11 is regulated by H2Aub meditated by deubiquitinase BAP1 [36]. This prompted us to investigate whether H2Aub contributes to the regulatory effects of EnP1 on SLC7A11 expression and ferroptosis. We speculated that H2Aub may function in a synergistic or complementary role in these processes. To explore this further, we assessed the binding affinity of H2Aub to the *SLC7A11* promoter under EnP1-overexpressing conditions. Our results showed that BAP1 overexpression reduces SLC7A11 expression, which was accompanied by a depletion of H2Aub (Fig 4A and 4B). Conversely, BAP1 knockdown elevates H2Aub accumulation, thereby enhancing SLC7A11 expression in host cells (Fig 4C and 4D).

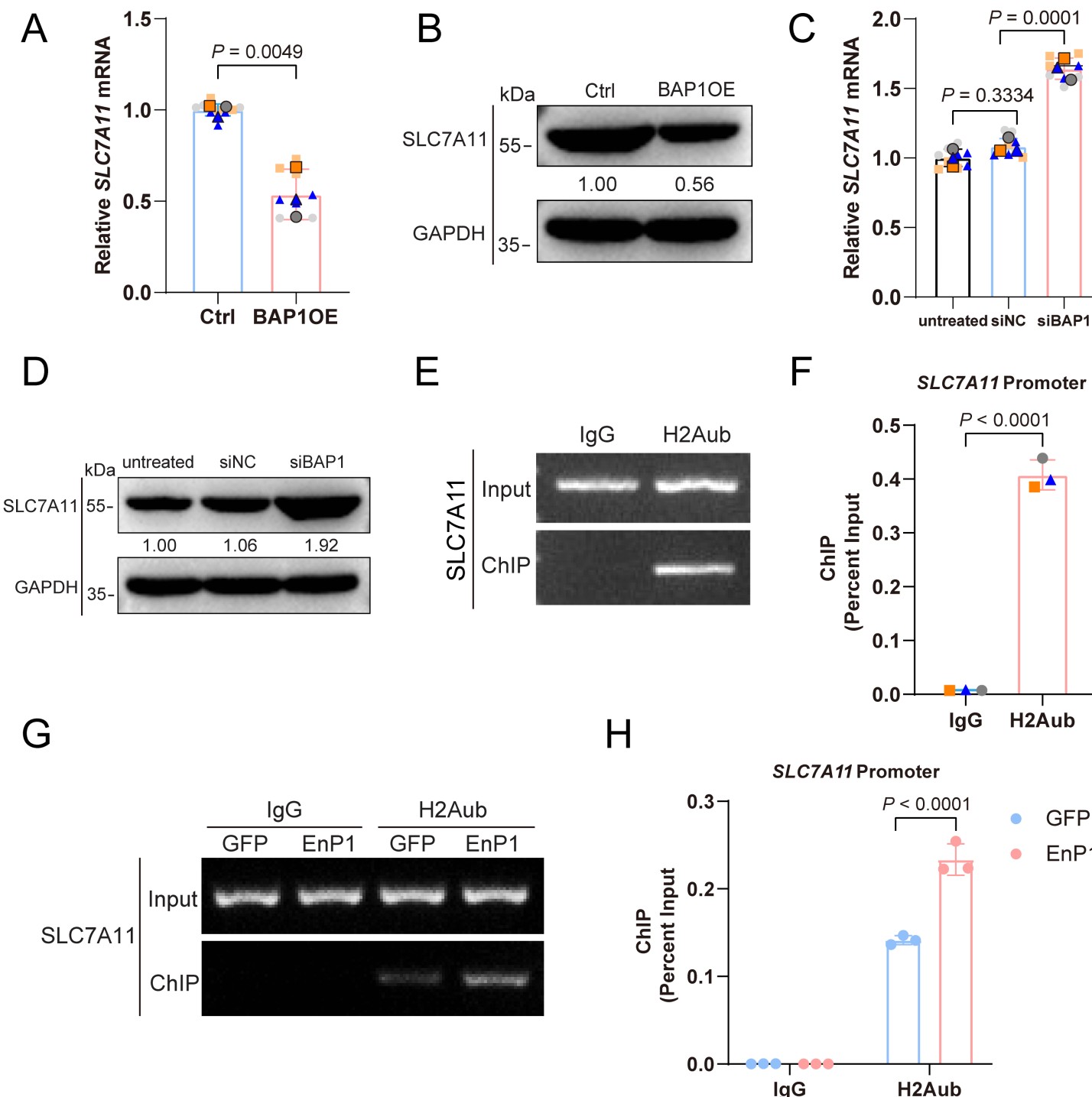

**Fig 4. EnP1 enhances the binding of H2Aub to the *SLC7A11* promoter. (A)** qRT-PCR analysis of SLC7A11 transcript levels in BAP1-overexpressing host cells showed consistent results across three independent biological replicates. **(B)** Immunoblot analysis of SLC7A11 levels following BAP1 overexpression. **(C)** qRT-PCR analysis of SLC7A11 transcript levels in BAP1-knockdown host cells showed consistent results across three independent biological replicates. **(D)** Immunoblot analysis of SLC7A11 levels after BAP1 knockdown. **(E)** ChIP-PCR assessment of H2Aub's interaction with the *SLC7A11* promoter. **(F)** ChIP-qPCR confirmation of the specificity of H2Aub binding to the *SLC7A11* promoter. **(G-H)** Following EnP1 overexpression, alterations in the binding affinity of H2Aub to the *SLC7A11* promoter were evaluated via ChIP-PCR (G) and ChIP-qPCR (H), providing evidence that EnP1 augments the affinity between H2Aub and the promoter. Anti-H2Aub rabbit mAb was used for IP, with rabbit IgG serving as a negative control to exclude nonspecific binding.

To elucidate the underlying epigenetic regulatory mechanism by which EnP1 modulates SLC7A11 via H2Aub, we next assessed the binding capacity of H2Aub to the *SLC7A11* promoter. Chromatin Immunoprecipitation (ChIP)-PCR (Fig 4E) and ChIP-qPCR (Fig 4F) results demonstrated that, compared to the control group, H2Aub is capable of binding to the promoter region of the *SLC7A11* gene. Subsequently, in EnP1 stable cell lines, further validation using ChIP-PCR (Fig 4G) and ChIP-qPCR (Fig 4H) confirmed that EnP1 significantly enhanced the binding of H2Aub to the *SLC7A11* promoter region. Collectively, these findings suggest that EnP1 upregulates H2Aub in host cells by modulating BAP1 expression. This, in turn, promotes the accumulation of H2Aub at the *SLC7A11* promoter, ultimately serving as a critical mechanism for the upregulation of SLC7A11.

### EnP1 mediates the regulation of host cell ferroptosis through H2Aub

Given that EnP1 directly regulates BAP1 expression in host cells, and that BAP1 controls H2Aub levels—subsequently influencing SLC7A11 expression—it is essential to determine whether H2Aub or BAP1 plays a central role in EnP1-mediated suppression of ferroptosis. To investigate this, we first assessed the levels of cellular reactive oxygen species (ROS) under both infection and EnP1 overexpression conditions using flow cytometry. In infected host cells, we observed a significant reduction in total cellular ROS levels when compared to uninfected controls, as indicated by the fluorescent probe DCFH-DA (Fig 5A and 5B). Notably, this reduction was further enhanced in EnP1 stable cell lines (Fig 5C and 5D), providing additional evidence that both EnP1 and microsporidian infection contribute to alleviating oxidative damage induced by ROS in host cells. Next, to investigate the role of BAP1 in mediating oxidative stress resistance, we over-expressed BAP1 and demonstrated its protective role against erastin-induced oxidative stress. This was evidenced by enhanced cell viability (Fig 5E) and attenuated LDH release (Fig 5F). Conversely, knocking down BAP1 compromised the host cells' ability to resist ferroptosis (Fig 5G and 5H). To further dissect the involvement of the H2Aub-SLC7A11 axis in EnP1-mediated ferroptosis resistance, we overexpressed BAP1 in EnP1 stable cell lines. The results revealed a significant reduction in SLC7A11 expression (Figs 5I and S4), and notably, the resistance of EnP1 stable cells to ferroptosis was significantly diminished (Fig 5J-5L). These results suggest that the EnP1's functional effects in ferroptosis suppression are dependent on H2Aub-mediated regulation.

This study elucidates the molecular mechanism by which the effector EnP1 modulates host cell ferroptosis resistance through H2Aub-mediated epigenetic regulation. EnP1 specifically binds to host histone H2A and suppresses BAP1 expression, thereby attenuating H2A deubiquitination and leading to genome-wide H2Aub accumulation. Notably, the enrichment of H2Aub at the *SLC7A11* promoter region directly facilitates transcriptional activation, which in turn enhanced ferroptosis resistance to host cells. This process ultimately establishes a favorable intracellular microenvironment that supports pathogen proliferation (Fig 6).

### Discussion

Pathogen effectors establish intricate networks with hosts, perturbing diverse physiological functions. By modulating host signaling pathways, they subvert biological processes and manipulate immune responses, thereby optimizing pathogen persistence and shaping the trajectory of infection [4]. We are confident that EnP1 operates in a similar fashion, with its regulatory mechanisms in the host requiring thorough elucidation. Our study has confirmed that H2A and its monoubiquitination modification mediate the regulation of host cell ferroptosis by microsporidia, thereby further enriching the regulatory network of EnP1.

Microbial nucleus-targeted effectors exert multifaceted regulation on host epigenetic modifications, leading to dysregulation of gene expression [37]. For instance, the effectors MoHTR1 and MoHTR2, secreted by *Magnaporthe oryzae*, are translocated into the host cell nucleus where they interact with multiple nuclear targets, functioning as transcriptional repressors. This interaction downregulates the expression of host defense genes, thereby increasing the susceptibility of the host to rice blast disease [38]. Phytophthora secretes numerous CRN effector proteins that localize to the nucleus

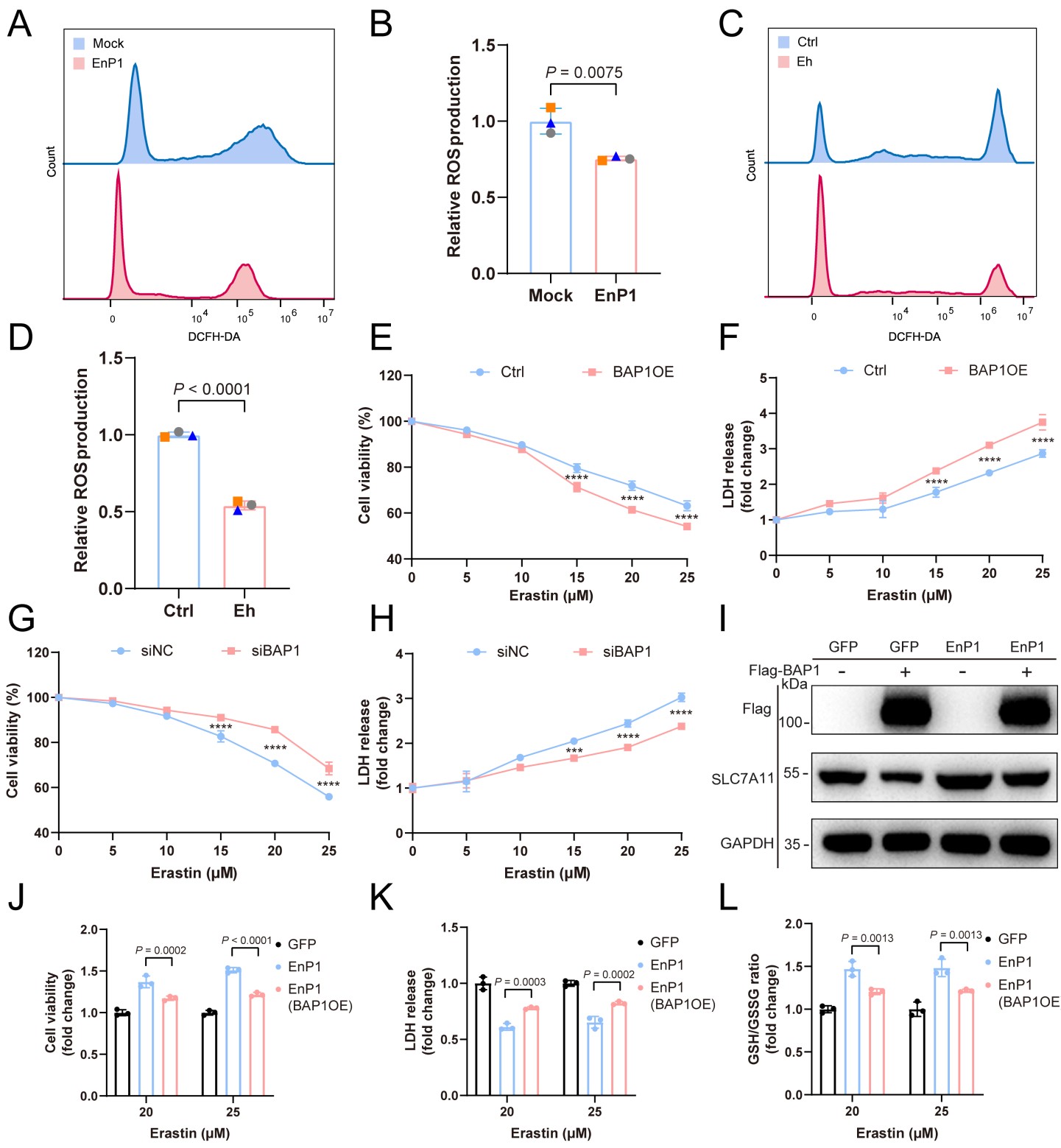

**Fig 5. The regulation of ferroptosis in host cells by EnP1 is partially contingent upon alterations in H2Aub levels. (A)** Flow cytometry analysis of reactive oxygen species content in EnP1 stably transfected cells versus control cells after Microsporidium infection. **(B)** Statistical analysis of fluorescence intensity from three independent experiments, including the one shown in (A). **(C)** Flow cytometry analysis of changes in reactive oxygen species

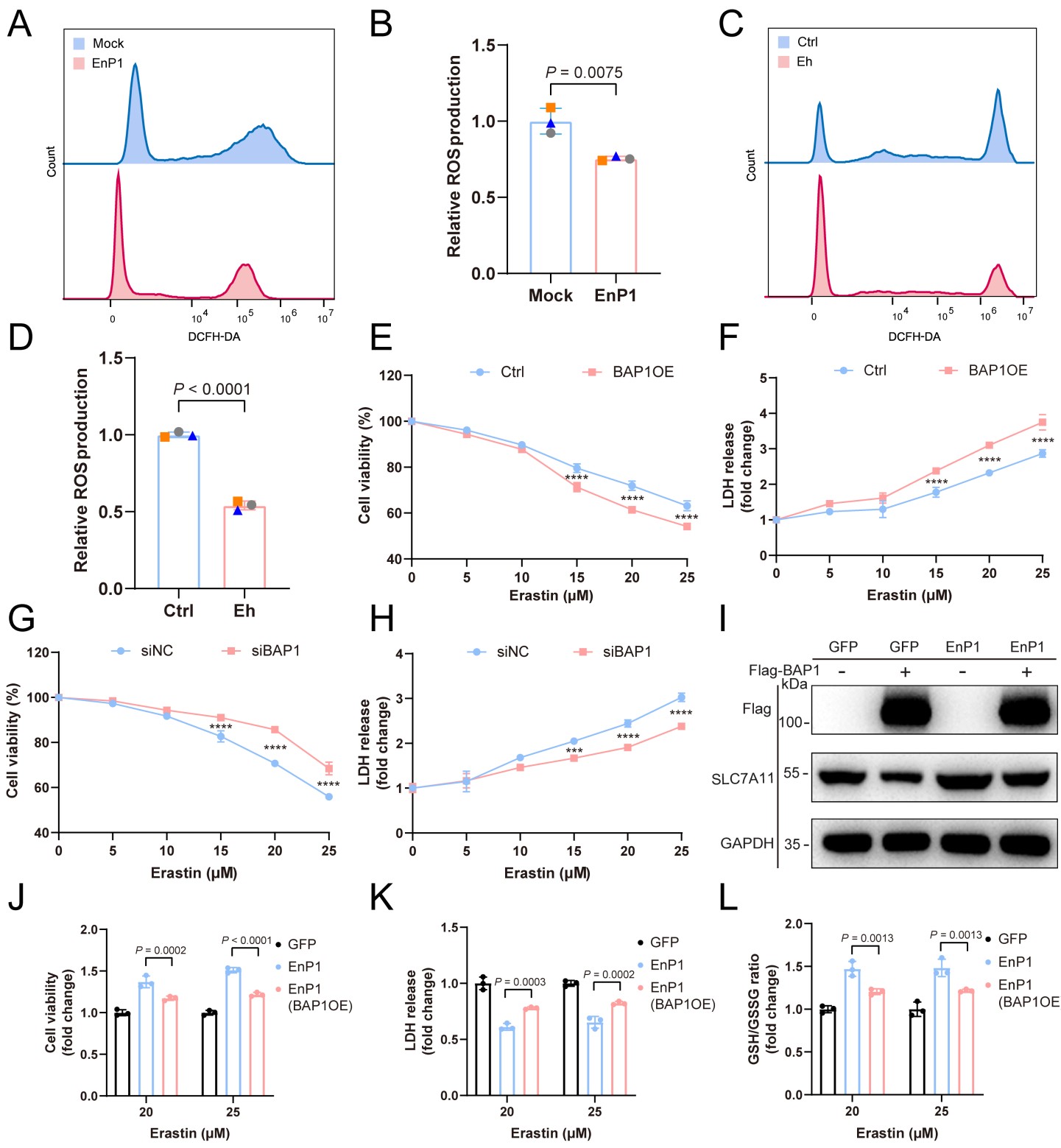

content in host cells following microsporidia infection. **(D)** Statistical analysis of fluorescence intensity from three independent experiments, including the one shown in (C). **(E)** Host cells overexpressing BAP1 (BAP1OE) or untreated were exposed to increasing concentrations of erastin, and their cell viability was assessed using CCK-8. **(F)** Host cells overexpressing BAP1 (BAP1OE) or untreated were treated with escalating concentrations of erastin, and LDH cytotoxicity was evaluated. **(G)** Host cells transfected with siBAP1 or siNC were exposed to increasing concentrations of erastin, and their cell viability was measured using CCK-8. **(H)** Host cells transfected with siBAP1 or siNC were treated with rising concentrations of erastin, and LDH cytotoxicity was determined. **(I)** Immunoblot analysis of SLC7A11 levels after BAP1 overexpression in EnP1-stabilized cell lines. **(J)** Cell viability of host cells expressing GFP, EnP1, and EnP1 with BAP1OE was assessed using CCK-8 after induction with 20 μM and 25 μM erastin. **(K)** LDH cytotoxicity assay of host cells expressing GFP, EnP1, and EnP1 with BAP1OE following induction with 20 μM and 25 μM erastin. **(L)** Measurement of the GSH to GSSG ratio in host cells expressing GFP, EnP1, and EnP1 with BAP1OE after induction with 20 μM and 25 μM erastin.

and exert cytotoxic effects [39]. Notably, CRN78 exhibits dual targeting to both membranes and the nucleus, where it interacts with various channel proteins to cooperatively suppress $H_2O_2$-mediated immune defenses [40]. In the case of microsporidia, *Nosema bombycis* can secrete the nucleus-targeted effector NbEBP1, which enters the host cell nucleus and regulates host gene expression, thereby affecting the cell cycle progression. Moreover, in the nematode-infecting microsporidian *Nematocida parisii*, at least four secreted effectors that target the nucleus of *Caenorhabditis elegans* have been identified. These proteins have been shown to interfere with host transcription and alter the intranuclear microenvironment [17]. Notably, the EnP1 protein secreted by Eh is widely distributed within the host nucleus, significantly increasing its potential to interact with multiple cellular targets. This suggests that its regulatory impact on host cell functions may be multifaceted, potentially playing an important role in processes such as the regulation of host cell ferroptosis.

The acidic patch is a highly conserved cluster of acidic amino acids on the surface of the histone H2A-H2B heterodimer, with H2Bub serving as its gatekeeper to regulate molecular access [41]. Pathogenic proteins can target the acidic patch, interacting with specific amino acid residues to induce chromatin state modifications that favor their replication [42]. A notable example is the KSHV LANA protein, which directly binds the H2A-H2B dimer to facilitate viral genome

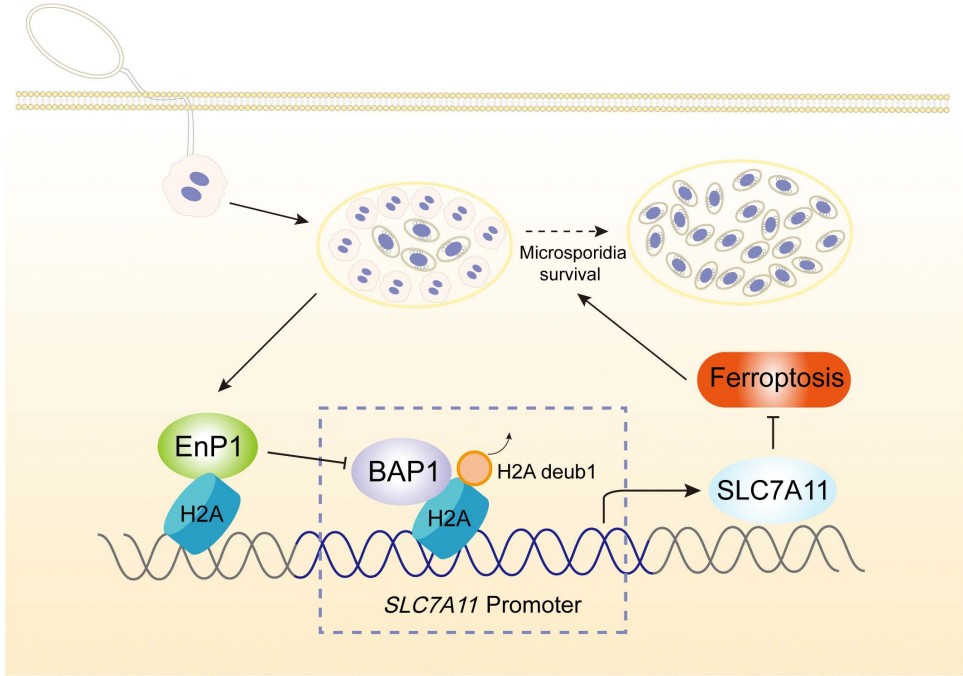

**Fig 6. Schematic illustrating how EnP1 facilitates microsporidian proliferation by modulating host cell H2Aub dynamics.**

maintenance and DNA replication [43,44]. The high-level expression of EnP1 throughout the entire lifecycle of microsporidia, coupled with its strong adhesion to the host cell surface, underscores its pivotal role in the infection process [17,45]. Our previous research has confirmed that EnP1 forms a complex with the host cell's H2B. Moreover, the mass spectrometry data from EnP1 pull-down experiments strongly suggest that EnP1 interacts with multiple targets within the host cell nucleus, with H2A being a highly probable secondary target [6]. Exploring whether EnP1 can bind to H2A has been both our primary motivation and a subject of keen curiosity. Moreover, there is a significant correlation between the modifications of H2A and the monoubiquitination levels of H2B [46].

The amino acid residues at the tail of histone H2A are modified through processes such as acetylation, methylation, phosphorylation, or ubiquitination, thereby regulating chromatin structure and gene expression [47]. Approximately 10% of H2A protein undergoes monoubiquitination, with key ubiquitination sites including K119, K13, and K15. The modification of K13/K15 is mediated by RNF168 and is primarily associated with DNA damage repair mechanisms [31,48–50]. Monoubiquitination of H2AK119 plays a pivotal role in the host's defense against pathogenic invasion. For instance, hepatitis C virus (HCV) infection leads to the degradation of RNF2, resulting in a reduction of H2AK119 monoubiquitination levels in host cells, thereby inducing the expression of HOX genes and facilitating viral propagation [51,52]. During microsporidia infection, we observed elevated H2Aub levels that appear to create a cellular environment favorable for parasite propagation. Notably, acetylation levels at two critical histone sites (H2AK5 and H2AK9) remained unchanged, while potential alterations in other modifications, such as phosphorylation, present intriguing avenues for future investigation.

H2Aub is primarily mediated by RING1B, a core component of the PRC1 [53], while its removal is catalyzed by BAP1, the core enzyme of the PR-DUB complex [54,55]. As an E3 ligase, RING1B-mediated elevation of H2Aub exerts well-established transcriptional repression, including on SLC7A11 [56,57]. The dynamic regulation of H2A ubiquitination and deubiquitination is essential for proper control of target genes [58]. Conversely, BAP1 removes H2Aub to prevent excessive accumulation at Polycomb target sites and protects active genes from aberrant silencing [59]. However, BAP1 can also suppress specific target gene expression through H2Aub-dependent deubiquitination at corresponding promoter regions [60,61]. Mechanistic studies demonstrate that BAP1-mediated removal of H2Aub at the *SLC7A11* promoter through its deubiquitinase activity transcriptionally represses SLC7A11 expression, thereby suppressing cysteine import to promote lipid peroxidation accumulation and potentiate ferroptosis [32,62]. Therefore, while BAP1 diminishes and PRC1 augments the binding of H2Aub on the *SLC7A11* promoter, both BAP1 and PRC1 inhibit the expression of SLC7A11. This indicates that the dynamic regulation of H2Aub is critically essential for expressing SLC7A11 [56]. H2Aub is intricately linked to H3K27 trimethylation (H3K27me3) in epigenetic regulation, particularly in the processes of gene silencing and chromatin remodeling [63,64]. PRC1, by ubiquitinating H2A, impedes chromatin unwinding, thereby enhancing the silencing effect mediated by PRC2 through H3K27me3 [64,65]. It remains to be investigated whether EnP1, by inhibiting BAP1 and increasing the host cell's H2Aub levels, affects the downstream levels of H3K27me3 and its target genes. As a strictly intracellular parasitic pathogen, the overall epigenetic regulation exerted by microsporidia on host cells requires further in-depth exploration.

Moreover, when we overexpressed p53 in the EnP1 stable cell line to counteract its inhibitory effect on SLC7A11 expression, we found that the expression level of SLC7A11 did not return to normal. Host cells still exhibited resistance to ferroptosis [6]. Hence, it appears that the presence of this effect is closely associated with alterations in H2Aub under microsporidia infection. Notably, the global alteration of H2Aub in host cells exhibits pleiotropic effects, exerting genome-wide transcriptional impacts as evidenced by transcriptomic profiling of EnP1-overexpressing cells. While elevated H2Aub levels may contribute to BAP1 downregulation, the precise mechanistic basis for BAP1 suppression requires further investigation.

The novel EnP1 regulatory network we have uncovered confirms the cunning and multifaceted nature of pathogen effectors in modulating their host. EnP1 not only engages multiple targets within the host cell nucleus, but its regulation of ferroptosis is not solely reliant on p53 as an intermediary. By reducing the BAP1-mediated deubiquitination of H2A, EnP1

enhances the occupancy of H2Aub at the *SLC7A11* promoter, thereby promoting its expression. Although EnP1 differentially modulates monoubiquitination of H2A and H2B, both modifications epigenetically reprogram the host cell to create a microenvironment conducive to microsporidian proliferation. These mechanistically distinct yet functionally synergistic strategies collectively suppress host ferroptosis while promoting pathogen replication, illustrating the evolutionary sophistication of microsporidian host manipulation.

## Materials and methods

### Cells and parasite strains

Human foreskin fibroblasts (HFF, ATCC CRL-2522) and human embryonic kidney 293T cells (HEK-293T, ATCC CRL-11268) were cultured at 37°C in a 5% $CO_2$ environment in Dulbecco's Modified Eagle Medium (DMEM, Cellmax) supplemented with 10% (v/v) fetal bovine serum (FBS, CellMax) and 1% (v/v) penicillin-streptomycin (Solarbio). The EnP1 stable cell line constructed using the lentiviral transduction system, as used in this study, has been previously validated in our published data [6]. The parasite strain used in the experiments was *Encephalitozoon hellem* (ATCC 50504).

### Parasite culture

Human foreskin fibroblasts (HFF) used for infecting Eh were cultured in Dulbecco's Modified Eagle Medium (DMEM, Gibco) supplemented with 10% (v/v) fetal bovine serum (FBS) and penicillin-streptomycin, and maintained at 37°C in a 5% $CO_2$ environment. HFF cells were infected with Eh at a multiplicity of infection (MOI) of 10:1. Mature spores were purified from the culture supernatant of infected HFF cells using a 27-gauge needle, followed by filtration through a 5 μm pore-size filter (Millipore) to remove host cell debris. The spores were then concentrated by high-speed centrifugation. Spore counts were performed using a hemocytometer, with at least three counts per infection to calculate an average. The remaining spores were stored in sterile phosphate-buffered saline (PBS) at 4°C for total protein extraction from the microsporidia.

### Assessment of Eh proliferation

Specific primers targeting the Eh small subunit ribosomal RNA (SSU, GenBank accession number: L19070) were designed (see S1 Table), and its coding sequence (CDS) was cloned into an eukaryotic expression vector. To perform absolute quantitative qPCR, different concentrations of template DNA were obtained through serial dilutions for quantifying the copy number of Eh. Cells were seeded in 12-well or 6-well plates and infected with microsporidia at an MOI of 10. After 12 hours of infection, the medium was replaced and cells were washed three times gently with PBS. All samples from different treatments were extracted using a genomic DNA extraction kit (Magen) when the cell density was close to 100%. The extracted DNA was subjected to qPCR using SSU-specific primers as the template for absolute quantification. A standard curve was constructed, and each sample's Ct value was compared with the standard curve to determine the copy number of Eh. All pathogen copy number data were derived from three independent biological replicates, with each replicate using pathogen stocks from different batches and infection intervals exceeding 24 hours.

Additionally, purified high-activity mature spores were used to infect HEK-293T cells cultured in 6-well plates, with cells subjected to different treatments as per experimental needs (e.g., under overexpression or knockdown of BAP1). Forty-eight hours post-infection, cells were fixed with 4% v/v paraformaldehyde and washed three times with TBST. Subsequently, the cells were incubated with 0.01% v/v Calcofluor White (Sigma-Aldrich) in PBS for 10 minutes. After incubation, the cells were washed three times with PBS and observed under a fluorescence microscope (Zeiss) to count and measure the number and size of parasitophorous vacuoles (PVs) formed by the blue-fluorescing spores. At least three biological replicates were performed for each experiment, with 20 images taken per well. All experimental operations and data analysis were performed in a blinded manner (double-blind experiment). The number and area of PVs for each

treatment condition were quantitatively analyzed using Fiji software (https://imagej.net/software/fiji/), and interference from individual spores was excluded.

## Plasmid construction

All plasmids used in this study were constructed using the Seamless Cloning Kit (Beyotime) or KLD enzyme mix (New England Biolabs) unless otherwise specified. The coding sequences (CDS) of the genes were amplified and assembled into the corresponding plasmid backbones, with plasmid construction performed using the Gibson method [66]. All primers were synthesized by Shanghai Shenggong Biotechnology Co., Ltd. The high-fidelity enzyme used for amplification was from 2×Phanta Max Master Mix (Vazyme). All newly prepared parent cloning vectors were confirmed by sequencing (Sangon).

## Immunofluorescence

To detect the colocalization of EnP1 and H2A, HEK-293T cells were first seeded in 24-well plates and co-transfected with the pcDNA3.1::EnP1-Flag and pcDNA3.1-3HA::H2A vectors. After 48 hours of transfection, the medium was removed with PBS, and the cells were fixed with 4% v/v paraformaldehyde. After washing, the cells were permeabilized in TBST containing 0.1% v/v Triton X-100 for 25 minutes and blocked in TBST containing 5% w/v BSA. Then, cells were incubated overnight at 4°C with diluted mouse anti-HA monoclonal antibody (1:1000, Abmart) and diluted rabbit anti-Flag monoclonal antibody (1:200, ABclonal). After incubation, the cells were washed three times with TBST, and Alexa Fluor 488-conjugated anti-mouse IgG antibody (1:2000, Invitrogen) and Alexa Fluor 594-conjugated anti-rabbit IgG antibody (1:300, ZSGB-BIO) were added and incubated at room temperature for 1 hour. After washing three times, cells were stained with DAPI for 10 minutes in the dark. Finally, the cells were mounted in an anti-fade mounting medium and observed with a Zeiss LSM 980 laser scanning microscope.

## Protein extraction and western blot

For total cell protein extraction, RIPA lysis buffer (ThermoFisher) with protease inhibitors was used for cell lysis on ice. Nuclear protein extraction kits (Solarbio) were used to extract nuclear proteins (NPE) from transfected or infected cells, following the manufacturer's protocol. After protein concentration was measured using the BCA kit (Beyotime), SDS-PAGE sample buffer (NCM Biotech) was added, and the samples were boiled at 100°C for 5 minutes. The proteins were separated by SDS-PAGE, transferred to PVDF membranes, and blocked with 5% w/v non-fat milk in TBST. After blocking, the membranes were incubated with primary and secondary antibodies, including anti-HA tag, anti-Flag tag, anti-SLC7A11, anti-ubiquitinated histone H2A (Lys119), anti-H3, anti-BAP1, anti-RING1B, anti-GAPDH, anti-tubulin, anti-mouse IgG-HRP, and anti-rabbit IgG-HRP, all in 5% w/v BSA in TBST. Protein bands were visualized using chemiluminescence (Millipore), and densitometric analysis was performed using Fiji software.

## Immunoprecipitation

Under cold conditions, nuclear proteins were extracted from HEK293T cells co-transfected with Flag-EnP1 and HA-H2A using Lipofectamine 3000 (Invitrogen) and supplemented with protease inhibitors (MedChemExpress). Nuclear proteins from single-transfected cells (Flag-EnP1 or HA-H2A) served as a negative control. Then, pre-balanced anti-HA (MedChemExpress) or anti-Flag magnetic beads (MedChemExpress) were added and incubated at 4°C for 3 hours with gentle agitation. After incubation, the beads were washed seven times with PBS containing 0.1% NP40 (Beyotime) to remove the nonspecific binding. The beads were resuspended in a sample buffer, subjected to SDS-PAGE separation, and immunoblotted for detection.

To evaluate whether EnP1 interacts with BAP1, the same experimental methods and reagents were used for reciprocal verification. The results were visualized using chemiluminescence (Millipore) to confirm the interaction.

## Protein co-expression in *E. coli*

A recombinant *E. coli* was constructed for co-expressing EnP1 and H2A using the plasmid pRSFduet-1, which contains two multiple cloning sites and each of them is preceded by a T7 promoter and ribosome binding site [67]. The coding sequences of EnP1-HA and H2A-Flag were cloned into two independent expression frames of the dual-expression vector pRSFDuet at the multiple cloning sites (MCS), with EnP1 tagged with His. The successfully constructed dual-expression plasmids were verified by sequencing and then transformed into *Escherichia coli* Rosetta (DE3) (Qiagen). The bacteria were cultured at 37°C in LB medium containing 100 µg/ml ampicillin until an OD600 of 0.6-0.8 was reached. Then, 0.5 mM IPTG (Sangon) was added to induce protein expression for 4 hours at 37°C. The bacteria were harvested by centrifugation, resuspended in buffer A (20 mM Tris-HCl, pH 7.5, 200 mM NaCl) containing 1 mM PMSF (Solarbio), and disrupted by sonication. After centrifugation at 12,000 g for 10 minutes at 4°C, the supernatant was filtered to remove debris. The clarified supernatant was added to a pre-equilibrated HisPure Nickel-Nitrilotriacetic Acid (Ni-NTA) column (Sangon) and washed with 10 column volumes of buffer A and 10 column volumes of buffer B (20 mM Tris-HCl, pH 7.5, 200 mM NaCl, 50 mM imidazole). The target protein was eluted with 5 column volumes of buffer C (20 mM Tris-HCl, pH 7.5, 500 mM imidazole). The purity of the protein was assessed by SDS-PAGE and Coomassie blue staining, and whether H2A was co-purified with EnP1 was confirmed through immunoblotting using an anti-Flag antibody.

## Prediction of the EnP1-H2A complex using AlphaFold3

Employing AlphaFold3 [27] through its publicly accessible web server (https://alphafoldserver.com), the structure of the EnP1–H2A protein complex was predicted. The amino acid sequences of EnP1 (accession: EHEL_010690) and histone H2A (UniProt ID: P04908) were retrieved and input as entity1 and entity2, respectively, with both configured as protein entities. Prediction was performed continuously under default settings, yielding five structural models of the EnP1-H2A complex. Each model was provided alongside graphical outputs such as a per-residue confidence score plot (pLDDT heatmap) and a predicted aligned error (PAE) matrix, which together assess local prediction reliability and inter-domain alignment confidence. Result files were downloaded directly from the output interface, and the predicted structures in PDB format were analyzed using PyMOL (Version 2.5.4). A custom Python script was executed within PyMOL to identify interaction interfaces between EnP1 and H2A by calculating residue contact pairs and binding regions based on distance thresholds.

## Reverse transcription quantitative PCR (qPCR)

Total RNA was extracted from specific cells using the RNA Easy Fast Cell kit (Qiagen), following the manufacturer's instructions. cDNA synthesis was performed using the 5×HiScript III qRT SuperMix (Vazyme), and qPCR was conducted using 2×Cham Q SYBR qPCR Master Mix (Vazyme) on a Bio-Rad CFX96 Real-Time PCR system. GAPDH was used as the internal control for the normalization of relative expression levels. The primer information is provided in S2 Table.

## RNA interference

Specific siRNA against BAP1 and control siRNA (siNC) were designed and synthesized by Sigma-Aldrich. To induce transient gene silencing, siRNA duplexes were transfected into HEK-293T cells in 6-well plates using Lipofectamine 3000 (ThermoFisher). The transfection protocol followed the manufacturer's guidelines, ensuring cells were in the logarithmic growth phase before transfection. After transfection, the cells were incubated for 24 hours to ensure effective uptake of siRNA and initiation of gene silencing. After 24 hours, Eh was added for infection, or ferroptosis inducer erastin

(MedChemExpress) was added, and changes in gene expression, cell condition, and microsporidia proliferation were observed.

At 48 hours post-transfection, cells were harvested for reverse transcription qPCR analysis of the mRNA expression levels of relevant host genes (e.g., BAP1, SLC7A11). At 72 hours post-transfection, cells were harvested for protein extraction and immunoblotting to evaluate the effects of siRNA-induced gene silencing on protein levels.

### Flow cytometry detection

DCFH-DA (Beyotime) was diluted 1:1000 in DMEM to a final concentration of 10 µM. The medium from HEK293T cells infected with Eh for 48 hours was removed, and the diluted DCFH-DA was added to the cells. After incubating at 37°C for 20 minutes, the cells were washed three times with PBS to remove any unincorporated DCFH-DA. Cells were collected and resuspended as a single-cell suspension, then analyzed for fluorescence intensity by flow cytometry using FITC settings. Data were analyzed using FlowJo software (https://www.flowjo.com/).

### Chromatin immunoprecipitation (ChIP)

1% formaldehyde (Invitrogen) was used to crosslink EnP1 stable transfected cells at 37°C, followed by 0.125 M glycine to terminate the crosslinking reaction. Cells were lysed with ChIP lysis buffer (Biosharp) containing Protease Inhibitor Cocktail (MedChemExpress) and subjected to micrococcal nuclease (Beyotime) digestion and sonication to fragment the DNA. After centrifugation at 10,000 g at 4°C for 10 minutes, the supernatant was incubated overnight at 4°C with an H2Aub rabbit monoclonal antibody (Cell Signaling) or rabbit IgG (negative control), forming the H2Aub-DNA complex. Protein A/G magnetic beads (ABclonal) were added to each tube and incubated at 4°C for 2 hours with gentle rotation. After removing the supernatant, the beads were thoroughly washed with washing buffer. DNA was eluted twice with elution buffer (1% SDS, 0.1M NaHCO3) and crosslinking was reversed at 65°C in 0.2 M NaCl for 2 hours. RNA contamination was removed using RNase A, and DNA was purified using phenol-chloroform extraction. The binding of SLC7A11 to H2Aub was determined by PCR and qPCR using specific primers for the *SLC7A11* promoter region.

### Cell viability assay

Equal amounts of cells were seeded in 96-well plates. After treatment with different concentrations of erastin (MedChemExpress) for 24 hours, 10 µl of CCK-8 reagent (100 µl total culture medium per well) was added to each well. After incubating at 37°C for 2 hours, the absorbance at 450 nm was measured using a microplate reader. The difference in absorbance between the two groups was compared to assess cell status after erastin induction.

To further evaluate membrane damage, lactate dehydrogenase (LDH) cell cytotoxicity detection kits (Beyotime) were used, and the absorbance at 490 nm was measured as per the kit instructions. For cells transfected with siRNA or overexpression plasmids, these steps were performed 48 hours post-transfection.

### Measurement of GSH and GSSG

Cells from different experimental and control groups were collected and washed thoroughly with PBS. The GSH/GSSG content was measured according to the instructions of the GSH/GSSG Assay Kit (Beyotime), and a standard curve was created using standard solutions. The absorbance of the samples was measured at 412 nm using a microplate reader. The GSH and GSSG levels in the cell lysates were calculated based on the standard curve and the relevant formula. For cells transfected with siRNA or overexpression plasmids, these steps were performed 48 hours post-transfection.

### Statistical analysis

All data were collected and analyzed without blinding. Data are presented as mean ± standard deviation (SD). Statistical analysis was performed using Prism (GraphPad) software. A comparison between the two groups was done

using a two-tailed unpaired Student's t-test; differences between multiple groups for a single variable were analyzed using one-way analysis of variance (ANOVA) followed by Tukey's multiple comparisons test. For comparisons involving multiple groups and multiple variables, two-way ANOVA was used with Sidak's multiple comparisons test. In this study, significance levels were defined as $*p < 0.05$, $**p < 0.01$, $***p < 0.001$, $****p < 0.0001$. Biological replicates for each statistical experiment were performed with at least three repetitions. In each graph, distinct colors and symbols represent different biological replicates. The corresponding technical replicates are displayed in lighter shades of the same color beneath their respective symbols. The statistical analysis of PVs was carried out using a strict double-blind method.

## Supporting information

**S1 Fig. EnP1 interacts with host H2A.** (A) Schematic illustrating the immunoprecipitation of nuclear proteins from EnP1-HA stably transfected cells using anti-HA magnetic beads. Mass spectrometry analysis revealed that the abundance of H2A was second only to H2B. (B) Immunoblot validation of EnP1 expression in EnP1-stably expressing cell lines. (C) The predicted structure of the EnP1-H2A complex was used in AlphaFold3, with EnP1 depicted in green, H2A in yellow, and the interacting regions highlighted in red. (D) Predicted Aligned Error (PAE) plot for the EnP1 (EHEL_010690)-H2A (UniProt: P04908) complex structure prediction. (E-F) Coomassie Brilliant Blue staining of proteins purified from the prokaryotic co-expression of H2A and EnP1 (E), with the upper and lower boxes indicating EnP1 and the pulled-down H2A, respectively; Coomassie Brilliant Blue staining of His-tag-free H2A purified from prokaryotic expression (F), with the box showing that while H2A is induced for expression, it cannot be pulled down in the absence of EnP1. Samples 1 and 2 represent uninduced and induced bacterial lysates, and samples 3–6 show proteins sequentially eluted using 500 mM imidazole.
(TIF)

**S2 Fig. EnP1 promotes H2Aub modification in host cells by restraining BAP1 expression.** (A) Quantification of H2A levels in EnP1-expressing host cells, with GFP-expressing stable cells serving as the negative control. (B) Quantification of H2A levels in host cells following microsporidia infection, with uninfected cells serving as the negative control. (C) Quantification of nuclear H2AK5ac levels in EnP1-expressing host cells relative to GFP-expressing controls. (D) Quantification of nuclear H2AK5ac levels in host cells following microsporidia infection, with uninfected cells serving as negative control. (E) Quantification of nuclear H2AK9ac levels in EnP1-expressing host cells relative to GFP-expressing controls. (F) Quantification of nuclear H2AK9ac levels in host cells following microsporidia infection, with uninfected cells serving as negative control. (G) qRT-PCR analysis of the alteration in RING1A expression in host cells stably expressing EnP1. (H) qRT-PCR analysis of the change in RING1A expression in host cells infected with microsporidia. (I-J) Immunoblotting analysis of RING1A levels in EnP1-expressing cells (I) and quantification of relative band intensities from three independent biological replicates using ImageJ software (J). (K-L) Immunoblot analysis of RING1A levels in host cells infected with microsporidia (K) and quantification of relative band intensities from three independent biological replicates using ImageJ software (L). (M) qRT-PCR analysis of the alteration in USP16 expression in host cells stably expressing EnP1. (N) qRT-PCR analysis of the change in USP16 expression in host cells infected with microsporidia. (O-P) Immunoblotting analysis of USP16 levels in EnP1-expressing cells (O) and quantification of relative band intensities from three independent biological replicates using ImageJ software (P). (Q-R) Immunoblot analysis of USP16 levels in host cells infected with microsporidia (Q) and quantification of relative band intensities from three independent biological replicates using ImageJ software (R). Quantification of mRNA and protein expression for each target was performed with three biologically independent replicates to ensure experimental reproducibility.
(TIF)

**S3 Fig. The level of H2Aub in host cells is positively correlated with microsporidia proliferation.** (A) Following initial Eh infection (MOI = 10) and subsequent washing, microsporidia parasite load was quantified by qPCR to validate washing efficiency and methodological feasibility. (B) After a 12-hour Eh infection (MOI = 10) followed by washing, microsporidia parasite load was quantified by qPCR to verify infection efficiency and methodological feasibility. (C) Immunoblot analysis of BAP1 expression in host cells using an αFlag antibody. (D) Cell viability was assessed using CCK-8 assay following BAP1 overexpression. (E) Cytotoxic effects of BAP1 overexpression were evaluated by measuring LDH release. (F) qRT-PCR analysis of BAP1 transcription levels in host cells after BAP1 knockdown. siNC: the negative control. (G) Immunoblot analysis of BAP1 protein levels in host cells after BAP1 knockdown. (H) Cell viability was assessed using the CCK-8 assay following BAP1 knockdown. (I) Cytotoxic effects of BAP1 knockdown were evaluated by measuring LDH release.
(TIF)

**S4 Fig. The regulation of ferroptosis in host cells by EnP1 is partially contingent upon alterations in H2Aub levels.** Band intensities from three independent biological replicates were quantified using ImageJ software (representative data shown in Fig 5I). The consistent results demonstrate that EnP1-mediated regulation of SLC7A11 is partially dependent on BAP1 expression levels.
(TIF)

**S1 Table. Mass spectrometry-based profiling of EnP1 protein interactome.**
(DOCX)

**S2 Table. List of primers for qPCR.**
(DOCX)

## Acknowledgments

We thank Meiling Wu from the Translational Medicine Core Facility of Shandong University for the consultation and instrument availability that supported this work.

## Author contributions

**Conceptualization:** Jingyu Guan, Hongnan Qu, Bing Han.

**Data curation:** Jingyu Guan, Yongliang Wang, Bing Han.

**Formal analysis:** Jingyu Guan, Ming Fu, Bing Han.

**Funding acquisition:** Jingyu Guan, Hongnan Qu, Bing Han.

**Investigation:** Jingyu Guan, Yongliang Wang, Ming Fu, Liyuan Tang, Musa Makongoro Sabi, Huimin Zhu, Bing Han.

**Methodology:** Jingyu Guan, Hongnan Qu, Bing Han.

**Project administration:** Huimin Zhu, Hua Cong, Chunxue Zhou, Hongnan Qu, Bing Han.

**Resources:** Huimin Zhu, Hua Cong, Chunxue Zhou, Hongnan Qu, Bing Han.

**Supervision:** Hongnan Qu, Bing Han.

**Validation:** Jingyu Guan, Hongnan Qu, Bing Han.

**Visualization:** Jingyu Guan, Hongnan Qu, Bing Han.

**Writing – original draft:** Jingyu Guan, Bing Han.

**Writing – review & editing:** Huimin Zhu, Hua Cong, Chunxue Zhou, Bing Han.

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
