## [Decision Letter · Decision Letter 0]

5 Nov 2025

EnP1 exploits H2Aub-dependent epigenetic reprogramming to promote microsporidia proliferation in host cells

PLOS Pathogens

Dear Dr. Han,

Thank you for submitting your manuscript to PLOS Pathogens. After careful consideration, we feel that it has merit but does not fully meet PLOS Pathogens's publication criteria as it currently stands. Therefore, we invite you to submit a revised version of the manuscript that addresses the points raised during the review process.

We look forward to receiving your revised manuscript.

Kind regards,

Emily R. Troemel

Academic Editor

PLOS Pathogens

Margaret Phillips

Section Editor

Editor-in-Chief

PLOS Pathogens

orcid.org/0000-0003-2946-9497

Michael Malim

PLOS Pathogens

orcid.org/0000-0002-7699-2064

**Additional Editor Comments:**

For all bar graphs in the paper, please plot all the individual datapoints that gave rise to these graphs, together with annotating which of the three independent experimental replicates the datapoints are from. As per Reviewer #2's suggestion please reference SuperPlots (PMID: 32346721, Fig 1D, far right) as an example.

Furthermore, the co-IP/WB studies showing interaction between Flag-ENP1 and HA-H2A would benefit from additional negative controls. Specifically, can the authors exclude the possibility that the tags on these constructs are responsible for the interaction? Single-transfected cells do not seem to be an adequate control here: rather, showing a lack of association between e.g. Flag-GFP and HA-H2A, and a lack of association between Flag-ENP and HA-GFP would provide reassurance that the interaction shown is due to ENP1 and H2A themselves, and not their tags.

**Journal Requirements:**

At this stage, the following Authors/Authors require contributions: Jingyu Guan, Yongliang Wang, Ming Fu, Liyuan Tang, Musa Makongoro Sabi, Huimin Zhu, Hua Cong, Chunxue Zhou, Huaiyu Zhou, Hongnan Qu, and Bing Han. Please ensure that the full contributions of each author are acknowledged in the "Add/Edit/Remove Authors" section of our submission form.

2) Please amend your detailed Financial Disclosure statement. This is published with the article. It must therefore be completed in full sentences and contain the exact wording you wish to be published.

3) Please send a completed 'Competing Interests' statement, including any COIs declared by your co-authors. If you have no competing interests to declare, please state "The authors have declared that no competing interests exist". Otherwise please declare all competing interests beginning with the statement "I have read the journal's policy and the authors of this manuscript have the following competing interests"

**Reviewers' Comments:**

Reviewer's Responses to Questions

**Part I - Summary**

Reviewer #1: I am satisfied with the significant efforts made by the authors to allay any concerns. The manuscript is ready for publication.

Reviewer #2: Overall, the authors seem to have made an effort to respond to reviewer comments, and I think in several cases this has clarified things or resolved a concern. In other cases, while the authors responses make sense on some level, I still have trouble understanding the results. For example, regarding my comment on Fig 3B (which I mentioned in my review as a specific example, but reflects a question about many of the figures): Given all to potential sources of variation between biological replicates, I find the size of the error bars (very small) and the statistical significance of this difference (p < 0.01) to be surprising. Parasites were from 3 separate batches, so I would expect some error in parasite counts between replicates, and variation in parasite infectivity/viability. I would also expect error/variation in host cell density plating on different days, small differences in host cell growth rates and viability, etc. There is presumably variation in washing away parasites after infection. Then variation in the read-out of parasite numbers at the end of the experiment. As a check to see if my intuition was fooling me, I tried simulating data for 3 replicates that would give similar means and standard deviations to what is depicted in Fig 3B (see attached figure). To get a mean of roughly 1.3x10^7 with a standard deviation of ~0.05 in the control (my estimates based on the author's figure; I assume the error bars represent standard deviations, though I can't find this in the legends), one might measure something like 1.25, 1.30 and 1.35 x10^7 in the 3 different biological replicates. For a technical replicate, I think this would already be pretty high reproducibility; given all the sources of variation I mentioned above, such minimal variation between complex biological replicates would seem to be very difficult to achieve. I'm not sure if there is something else I am missing? I strongly suggest displaying this kind of data using SuperPlots (PMID: 32346721, e.g., Fig 1D, far right), which show all of the individual measurements alongside the mean and standard (and where applicable, color-code the technical replicates within each biological replicate).

Because of these sorts of issues, it is unclear to me what to make of the data. And given the relatively small effects in many cases, it is also unclear to me how important the reported differences are parasite replication, and consequently, how impactful this manuscript will be on the field.

Reviewer #3: The authors addressed every weakness/issue/concern I noted in my previous review.

**Part II – Major Issues: Key Experiments Required for Acceptance**

Reviewer #1: (No Response)

Reviewer #2: (No Response)

Reviewer #3: (No Response)

**Part III – Minor Issues: Editorial and Data Presentation Modifications**

Reviewer #1: (No Response)

Reviewer #2: (No Response)

Reviewer #3: (No Response)

PLOS authors have the option to publish the peer review history of their article (what does this mean? ). If published, this will include your full peer review and any attached files.

**Do you want your identity to be public for this peer review?** For information about this choice, including consent withdrawal, please see our Privacy Policy .

Reviewer #1: No

Reviewer #2: No

Reviewer #3: No

**Figure resubmission:**

**Reproducibility:**



---

## [Editor Report · Decision Letter 1]

29 Dec 2025

Dear Prof. Han,

We are pleased to inform you that your manuscript 'EnP1 exploits H2Aub-dependent epigenetic reprogramming to promote microsporidia proliferation in host cells' has been provisionally accepted for publication in PLOS Pathogens.

Best regards,

Emily R. Troemel

Academic Editor

PLOS Pathogens

Dominique Soldati-Favre

Section Editor

PLOS Pathogens

Sumita Bhaduri-McIntosh

Editor-in-Chief

PLOS Pathogens

orcid.org/0000-0003-2946-9497

Michael Malim

Editor-in-Chief

PLOS Pathogens

orcid.org/0000-0002-7699-2064
---

## [Editor Report · Acceptance letter]

Dear Prof. Han,

We are delighted to inform you that your manuscript, " 

EnP1 exploits H2Aub-dependent epigenetic reprogramming to promote microsporidia proliferation in host cells," has been formally accepted for publication in PLOS Pathogens.

Best regards,

Sumita Bhaduri-McIntosh

Editor-in-Chief

PLOS Pathogens

orcid.org/0000-0003-2946-9497

Michael Malim

Editor-in-Chief

PLOS Pathogens

orcid.org/0000-0002-7699-2064